# Quantitative imaging of volcanic $SO_2$ plumes with Fabry Pérot Interferometer Correlation Spectroscopy

Christopher Fuchs[1], Jonas Kuhn[1,2], Nicole Bobrowski[1,2], and Ulrich Platt[1,2]

[1]Institute of Environmental Physics, University of Heidelberg, Germany
[2]Max Planck Institute for Chemistry, Mainz, Germany

**Correspondence:** C. Fuchs (cfuchs@iup.uni-heidelberg.de), J. Kuhn (jkuhn@iup.uni-heidelberg.de)

**Abstract.** We present first measurements with a novel imaging technique for atmospheric trace gases in the UV spectral range. Imaging Fabry Pérot Interferometer Correlation Spectroscopy (IFPICS), employs a Fabry Pérot Interferometer (FPI) as wavelength selective element. Matching the FPIs distinct, periodic transmission features to the characteristic differential absorption structures of the investigated trace gas allows to measure differential atmospheric column density (CD) distributions of numerous trace gases with high spatial and temporal resolution. Here we demonstrate measurements of sulphur dioxide (SO2) while earlier model calculations show that bromine monoxide (BrO) and nitrogen dioxide (NO2) are also possible. The high specificity in the spectral detection of IFPICS minimises cross interferences to other trace gases and aerosol extinction allowing precise determination of gas fluxes. Furthermore, the instrument response can be modelled using absorption cross sections and a solar atlas spectrum from the literature, thereby avoiding additional calibration procedures, e.g. using gas cells. In a field campaign, we recorded the temporal CD evolution of $SO_2$ in the volcanic plume of Mt. Etna with an exposure time of $1\,\mathrm{s}$ and $400 \times 400$ pixels spatial resolution. The temporal resolution of the time series was limited by the available non-ideal prototype hardware to about $5.5\,\mathrm{s}$. Nevertheless, a detection limit of $2.1 \times 10^{17}\,\mathrm{molec\,cm^{-2}}$ could be reached, which is comparable to traditional and much less selective volcanic $SO_2$ imaging techniques.

## 1 Introduction

Ground based imaging of atmospheric trace gas distributions has a great potential to give new insights into mixing processes and chemical conversion of atmospheric trace gases by allowing their observation at high spatio-temporal resolution. Whereas present space borne trace gas imaging provides daily global coverage with a spatial resolution of a few km (e.g. Veefkind et al., 2012), ground based observation can potentially reach a spatial resolution in the order of metres and a temporal resolution in the single digit Hz range. Such techniques in particular allow the investigation of trace gas distributions with strong gradients and short time scale chemical conversions.

There are several approaches for imaging trace gas distributions using scattered sunlight in the UV-Vis wavelength range (see e.g. Platt and Stutz, 2008; Platt et al., 2015): An image can be scanned pixel by pixel with a telescope and recorded spectra are evaluated to determine the trace gas column density (whiskbroom approach). Alternatively, with a more complex optics and

a two dimensional detector, one detector dimension of the spectrograph can be used for spatially resolving an image column. Column by column (or pushbroom) scanning then resolves an image. The high spectral resolution of the spectrograph based techniques allows the accurate and simultaneous identification of several trace gases, however, the light throughput and the scanning process severely limit the temporal resolution. A third approach applies tunable filters to resolve the trace gas spectral features, e.g. acousto-optical tunable filter (Dekemper et al., 2016), as wavelength selective elements for an entire image frame.

The application of tunable filters can have high spectral resolution and hence high trace gas selectivity, however, due to limited light throughput the temporal resolution lies in the order of minutes. A fourth imaging technique uses a small number (typically two) wavelength channels selected by static filters, e.g. interference filters (Mori and Burton, 2006). This approach can be quite fast with a temporal resolution in the order of seconds, the trace gas selectivity, however, strongly depends on the correlation of trace gas absorption with the wavelength selective elements and usually is rather marginal.

Fabry Pérot Interferometers (FPIs) exhibit a periodic spectral transmission pattern, which can be matched to periodic spectral features (typically due to rotational or vibrational structures of electronic transitions) of the trace gas absorption, thereby yielding very high correlation for some trace gases. Imaging Fabry Pérot Interferometer Correlation Spectroscopy (IFPICS) thus essentially combines the advantage of fast image acquisition with selective spectral identification of the target trace gas. IFPICS was proposed by Kuhn et al. (2014) and discussed in Platt et al. (2015) for volcanic $SO_2$. Kuhn et al. (2019) demonstrated the feasibility with a one-pixel prototype for volcanic $SO_2$ and evaluated its applicability to other trace gases.

Here we present first imaging measurements (at a resolution of $400 \times 400$ pixels, $1\,s$ exposure time) performed with IFPICS and confirm its high selectivity and sensitivity. A prototype instrument for $SO_2$ was tested at Mt. Etna volcano, Italy, showing a noise equivalent signal between $2.1 \times 10^{17} - 5.5 \times 10^{17}\,molec\,cm^{-2}\,s^{-1/2}$. Furthermore, we show that the instrument response can be modelled and thereby intrinsically calibrated, using a solar atlas spectrum and literature trace gas absorption cross sections.

Existing interference filter based $SO_2$ cameras used for e.g. the quantification of volcanic trace gas emission fluxes into the atmosphere (Mori and Burton, 2006; Bluth et al., 2007; Kern et al., 2015a), exhibit strong cross interferences to aerosol scattering extinction and other trace gases (Lübcke et al., 2013; Kuhn et al., 2014). Furthermore, these techniques require in field calibration. Besides the thereby induced systematic errors that propagate into the emission flux quantification, the detection limit is mostly determined by these cross interferences. Thus, the applicability of the technique is limited to strong emitters with respective plume and weather conditions. The much higher selectivity of IFPICS largely extends the range of applicable conditions (e.g. to ship emissions and weaker emitting volcanoes) and significantly reduces the systematic errors. Furthermore, the extension of the technique to other trace gases e.g. bromine monoxide (BrO), formaldehyde (HCHO) or nitrogen dioxide ($NO_2$) can give new important insights into short scale chemical conversion processes in the atmosphere.

## 2 Imaging Fabry Pérot Interferometer Correlation Spectroscopy (IFPICS)

Similarly to the $SO_2$ camera principle (e.g. Mori and Burton, 2006; Bluth et al., 2007), IFPICS uses an apparent absorbance (AA) $\tilde{\tau} = \tau_A - \tau_B$, i.e. the difference between two measured optical densities $\tau_A$ and $\tau_B$, to quantify the column density (CD)

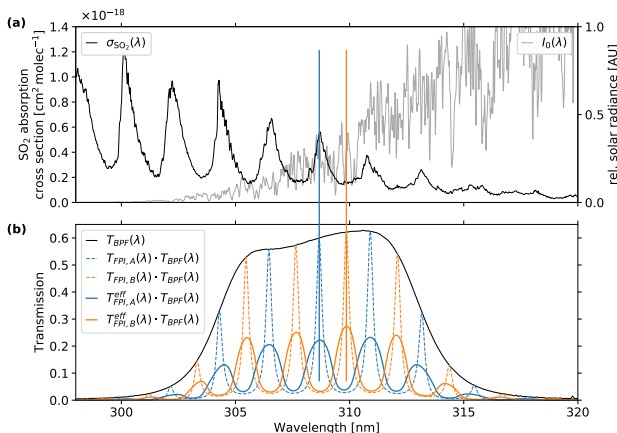

**Figure 1.** Spectral variation of: **(a)** The $SO_2$ absorption cross section $\sigma_{SO_2}$ (black drawn, left axis, according to Bogumil et al. (2003)) and the scattered skylight radiance $I_0(\lambda)$ (gray drawn, right axis in relative units), given by Eq. 2. **(b)** The FPI transmissions in settings A and B yielding the maximum AA detectable (best correlation/anti-correlation to $\sigma_{SO_2}$) in the spectral range specified by the used band pass filter (BPF). Shown are: The BPF transmission $T_{BPF}(\lambda)$ (black) and the FPI transmission spectrum for a single beam approach according to Eq. 6 in on-band $T_{FPI,A}(\lambda)$ (dashed blue, correlation with $\sigma_{SO_2}$) and off-band $T_{FPI,B}(\lambda)$ setting (dashed orange, anti-correlation with $\sigma_{SO_2}$). The effective FPI transmission spectrum including an incident angle distribution according to Eq. 7 in on-band $T_{FPI,A}^{eff}(\lambda)$(drawn blue) and off-band $T_{FPI,B}^{eff}(\lambda)$ setting (drawn orange).

$S = \int_0^L c(l)\,dl$, i.e. the integrated concentration $c$ of the measured gas along a light path $L$ for each pixel of the image. The AA is calculated from two (or more) spectral settings that yield a maximum correlation difference to the gas absorption spectrum. Ideally the periodicity of the FPI fringes are matched to periodic spectral absorption features as shown in Fig. 1 for $SO_2$. For IFPICS we use two spectral settings A and B. Setting A exhibits on-band absorption, where the FPI transmission maxima coincide with the $SO_2$ absorption maxima and hence correlating with the differential absorption structures of $SO_2$. Setting B, uses an off-band position where the FPI transmission maxima anti-correlate with the differential $SO_2$ absorption structures (see Fig. 1). The spectral separation between setting A and B is thereby reduced by a factor of $\approx 30$ (in the case of $SO_2$) to only $\approx 0.5\,\mathrm{nm}$ in contrast to $\approx 10 - 15\,\mathrm{nm}$ for traditional $SO_2$ cameras (see Lübcke et al., 2013; Kern et al., 2015a), which minimises broad band interferences due to e.g. scattering and extinction by aerosols or other absorbing gases. This application of an FPI is similar to approaches reported by Wilson et al. (2007) and Vargas-Rodríguez and Rutt (2009), for the detection of carbon monoxide, carbon dioxide and methane in the infrared spectral range.

By measuring the optical density $\tau_A = \ln(I_A/I_{0,A})$ and $\tau_B = \ln(I_B/I_{0,B})$ in both spectral settings A and B respectively, the relation between the AA $\tilde{\tau}(S)$ with the CD $S$ is given by

$$\tilde{\tau}(S) = \tau_A - \tau_B = -\log\frac{I_A}{I_{0,A}} + \log\frac{I_B}{I_{0,B}} = k(S) = \Delta\tilde{\sigma}(S)\cdot S, \tag{1}$$

where $I_A$, $I_B$ denote the radiances with and $I_{0,A}$, $I_{0,B}$ the radiance without the presence of the target trace gas in the absorption light path. The absorber free reference radiances $I_{0,A}$ and $I_{0,B}$ can be determined from e.g. a reference region within

the image. The differential weighted effective trace gas absorption cross section $\Delta\tilde{\sigma}(S)$ becomes independent of $S$ for small AAs ($\tilde{\tau} \ll 1$). At higher AAs saturation effects occur due to the non-linearity of Lambert-Beer's law, however knowledge of the absorption cross sections, the background radiation spectrum, and the instrument transmission allows to calculate $\tilde{\tau}$ for arbitrary CDs $S$ using a numerical model.

## 2.1 Instrument model

The AA $\tilde{\tau}$ is modelled for given target trace gas CDs $S$ by simulating the incoming radiances $I_A$, $I_B$ and $I_{0,A}$, $I_{0,B}$. As incident radiation a high-resolution, top of atmosphere (TOA) solar atlas spectrum $I_{0,TOA}(\lambda)$ is used according to Chance and Kurucz (2010). The TOA spectrum is scaled by the wavelength $\lambda^{-4}$ approximating a Rayleigh scattering atmosphere. Since our measurement wavelength range, of 304-313 nm for $SO_2$, overlaps with absorption of ozone ($O_3$), the TOA spectrum is corrected for the stratospheric $O_3$ absorption by multiplying all intensities with the Lambert-Beer's term $e^{-\sigma_{O_3}(\lambda)\cdot S_{O_3}}$. Where $S_{O_3}$ denotes the total atmospheric ozone slant column density, e.g. according to TEMIS Database, (Veefkind et al., 2006), and $\sigma_{O_3}$ the $O_3$ absorption cross section according to Serdyuchenko et al. (2014). This yields the scattered skylight radiance $I_0(\lambda)$

$$I_0(\lambda) = I_{0,TOA}(\lambda) \cdot e^{-\sigma_{O_3}(\lambda)\cdot S_{O_3}} \cdot f(\lambda^{-4}), \tag{2}$$

indicated in Fig. 2. Based on $I_0(\lambda)$ the radiances measured by the instrument for the two respective spectral settings are calculated with the absorption of trace gases and the spectral instrument transfer function $T_{instr}(\lambda)$. The investigated target trace gas $j$ (in this work $SO_2$) and potentially interfering trace gas species $k$ (in this work $O_3$) are added according to Lambert-Beer's law. In the following we use the index $i$, denoting the FPI settings A and B, respectively. The quantity $I_{0,i}$ thereby denotes the reference radiance excluding the target trace gas $j$ from the light path (see Fig. 2).

$$I_i = \int d\lambda \, I_0(\lambda) \cdot \exp\left(-\sigma_j(\lambda)S_j - \sum_k \sigma_k(\lambda)S_k\right) \cdot T_{instr,i}(\lambda) \tag{3}$$

$$I_{0,i} = \int d\lambda \, I_0(\lambda) \cdot \exp\left(-\sum_k \sigma_k(\lambda)S_k\right) \cdot T_{instr,i}(\lambda) \tag{4}$$

The spectral instrument transfer functions $T_{instr,i}(\lambda)$ for the two spectral settings

$$T_{instr,i}(\lambda) = T_{FPI,i}^{eff}(\lambda) \cdot T_{BPF}(\lambda) \cdot Q(\lambda) \cdot \eta(\lambda) \tag{5}$$

consists of the measured band pass filter (BPF) transmission spectrum $T_{BPF}(\lambda)$, the spectral (i.e. wavelength dependent) quantum efficiency $Q(\lambda)$ of the detector, and a spectral loss factor $\eta(\lambda)$ of the employed optical components (e.g. by reflections). Considering only a single, parallel beam of light traversing the instrument the FPI transmission spectrum $T_{FPI,i}(\lambda)$ is defined by the Airy function (Perot and Fabry, 1899)

$$T_{FPI,i}(\lambda;\alpha_i,d,n,R) = \left[1 + \frac{4\cdot R}{(1-R)^2} \cdot \sin^2\left(\frac{2\pi\cdot d \cdot n \cdot \cos\alpha_i}{\lambda}\right)\right]^{-1} \tag{6}$$

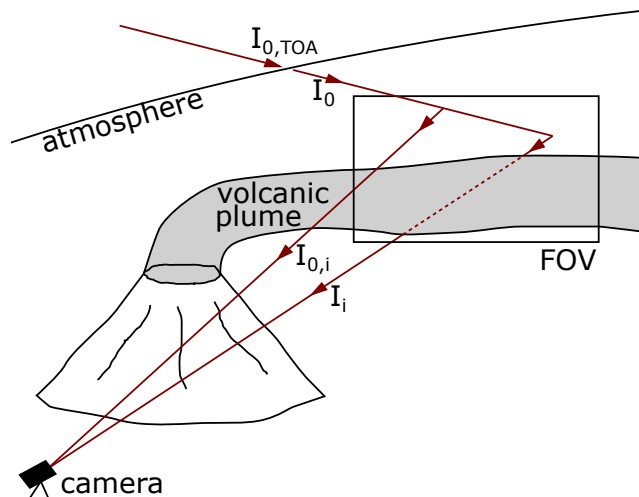

**Figure 2.** Schematic of the IFPICS measurement geometry including the simulated radiances used in the instrument model. The incident top of atmosphere (TOA) radiation $I_{0,TOA}$ is propagating through the atmosphere and is potentially scattered into the IFPICS camera field of view (FOV) yielding the scattered skylight radiance $I_0$. The camera records radiation in the respective FPI settings $i = A$ and B that either traverses the volcanic plume $I_i$ or originates from a plume free area within the FOV $I_{0,i}$.

with the light beam incidence angle $\alpha_i$ for the two spectral settings onto the FPI, the FPI mirror separation $d$, the refractive index $n$ of the medium inside the FPI, and the FPI reflectivity $R$ (see Tab. 1, Fig. 1 and Fig. 3, (d)). The FPI used in this work

is static and air-spaced, meaning $d$, $n$, and $R$ are fixed. Hence, the incidence angle $\alpha_i$ is the exclusive free parameter available to tune the FPIs transmission spectrum $T_{FPI,i}$ between settings $i = A$ and $i = B$ respectively. The change in $\alpha_i$ is achieved by tilting the FPI optical axis with respect to the imaging optical axis (see Section 2.2).

However, in reality a spectral setting will always contain a range of incidence angles onto the FPI. In this work we assume cone shaped light beams, with half cone opening angles $\omega_c$, where the entire cone can be tilted by $\alpha_i$ relative to the normal of

the FPI mirrors (see Fig. 3, (d)). From this assumption follows that the incidence angles $\alpha_i$ are distributed over a cone with the incidence angle distribution $\gamma(\alpha_i, \omega_c, \vartheta, \varphi)$, where $\vartheta$ and $\varphi$ are the polar and azimuth angles, respectively. Hence, the single beam FPI transmission spectrum $T_{FPI,i}(\lambda)$ of Eq. 6 is extended by a weighted average over $T_{FPI,i}(\lambda; \gamma(\alpha_i, \omega_c, \vartheta, \varphi), d, n, R)$, giving the effective FPI transmission spectrum $T_{FPI,i}^{eff}(\lambda)$

$$T_{FPI,i}^{eff}(\lambda; \gamma(\alpha_i, \omega_c), d, n, R) = \frac{1}{N(\gamma(\alpha_i, \omega_c))} \int\limits_{0}^{\varphi_{max}} \int\limits_{\vartheta_{min}}^{\vartheta_{max}} T_{FPI,i}(\lambda; \gamma(\alpha_i, \omega_c, \vartheta, \varphi), d, n, R) \sin\vartheta \, d\vartheta \, d\varphi. \quad (7)$$

Thereby, $N(\gamma(\alpha_i, \omega_c))$ denotes the weighting function with $N(\gamma(\alpha_i, \omega_c)) = \int_0^{\varphi_{max}} \int_{\vartheta_{min}}^{\vartheta_{max}} \sin\vartheta \, d\vartheta \, d\varphi$ given by integral in Eq. 7 excluding the integrand $T_{FPI,i}$ itself, $\vartheta$ the polar angle and $\varphi$ the azimuth angle of the spherical integration within boundaries defined by the tilted cone shaped light beams. E.g.: for a non-tilted FPI ($\alpha_i = 0$) the integration boundaries are $\vartheta \in [0, \omega_c]$ and $\varphi \in [0, 2\pi]$, for a tilted FPI however, the transformation of $\gamma(\alpha_i, \omega_c, \vartheta, \varphi)$ is more complex and requires several case analyses.

The incidence angle distribution $\gamma(\alpha_i, \omega_c)$ will affect the shape of the FPI transmission spectrum by decreasing the effective finesse $F$ of the FPI leading to a blurring of the FPI fringes (see Fig. 1).

## 2.2 The IFPICS prototype

The IFPICS prototype is a newly developed instrument, designed to function under harsh environmental conditions in remote locations like e.g. in the proximity to volcanoes. Hence, the prototype is designed to be small with dimensions of $200\,\mathrm{mm} \times 350\,\mathrm{mm} \times 130\,\mathrm{mm}$, lightweight with 4.8 kg (see Fig. 3, (a)) and has a power consumption $< 10\,\mathrm{W}$, thus can be battery-operated for several hours. A 2D UV-sensitive CMOS sensor with $2048 \times 2048$ pixel resolution (SCM2020-UV provided by *EHD imaging*) is used to acquire images. The sensor is operated in $4 \times 4$ binning mode yielding a final image resolution of $512 \times 512$ pixel. However, we found that the software of the SCM2020-UV image sensor does not allow sufficiently precise triggering. Therefore $\approx 0.6$ seconds are lost in each image acquisition, which severely limits the operation of the IFPICS camera. Replacement of the sensor by a scientific-grade UV detector array will solve this problem in future studies.

The internal camera optics is highly modular and easily adjustable. The IFPICS prototype employs an image side telecentric optical setup as proposed in Kuhn et al. (2014, 2019). A photograph and a schematic drawing are shown in Fig. 3. An aperture and a lens (lens 1) parallelise incoming light from the imaging field of view (FOV) before it traverses the FPI and the BPF. A second lens (lens 2) focusses the light onto the 2D UV-sensitive sensor. Thereby, in good approximation, all the pixels of the image experience the same spectral instrument transfer function $T_{instr,i}(\lambda)$ for the two wavelength settings.

The FPI is the central optical element of the IFPICS prototype and is implemented as static air-spaced etalon with fixed $d$, $n$, and $R$ (provided by *SLS Optics Ltd.*). The mirrors are separated using ultra low expansion glass spacers to maintain a constant mirror separation $d$ and parallelism over the large clear aperture of $20\,\mathrm{mm}$ even in variable environmental conditions. In order to tune the spectral transmission $T_{FPI}^{eff}$ between setting A and B a variation of the incidence angle $\alpha$ is applied. The FPI can be tilted within the parallelised light path using a stepper motor. The stepper motor has a resolution of $0.9°$ per step, is equipped with a planetary gearbox (reduction rate 1/9) and operated in micro-stepping mode (1/16) resulting in a resolution of $0.00625°$ per motor step. The time required for tilting between our settings A and B is $\approx 0.15\,\mathrm{s}$. We favour the approach of tilting the FPI over changing internal physical properties like, e.g. the mirror separation $d$ by piezoelectric actuators, as it keeps simplicity, robustness, and accuracy high for measurements under non-laboratory conditions. However it needs to be considered, that the tilting of the FPI will generate a linear shift between the respective images acquired in setting A and B, requiring an alignment in the evaluation process.

The half cone opening angle $\omega_c$ is determined by the entrance aperture $a$ and the focal length $f$ of lens 1 and can be calculated by $\omega_c = \arctan(a/2f)$. The physical properties of the optical components and the instrument are listed in Tab. 1 and were mostly chosen according to the dimensioning assumed in the calculations of Kuhn et al. (2019).

The FPI design with fixed $d$, $n$ and $R$ (see Fig. 3, (c)) in particular is chosen to inherently generate a transmission spectrum matching the differential absorption structures of $SO_2$. This includes the basic idea that the untilted FPI ($\alpha_i = 0°$) already matches the on-band position A. In our case however, the manufacturing accuracy of $d$ lies within one free spectral range ($\approx 2\,\mathrm{nm}$ for $SO_2$) yielding that $\alpha_B = 0°$, corresponds to a off-band (B) position ($T_{FPI,B}^{eff}$) and the on-band (A) position ($T_{FPI,A}^{eff}$)

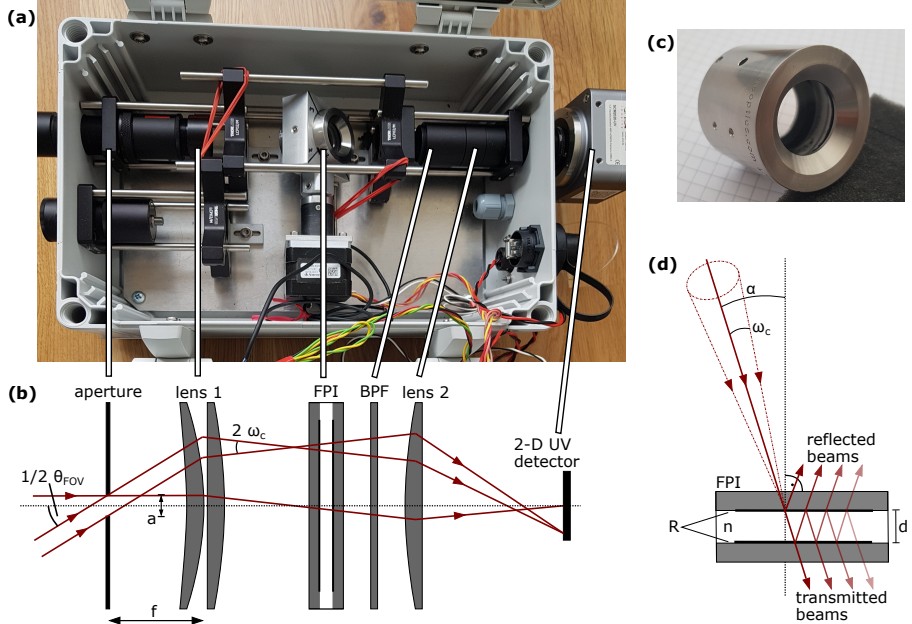

**Figure 3. (a)**: Photograph of the IFPICS instrument. The physical dimensions are $200\,\text{mm} \times 350\,\text{mm} \times 130\,\text{mm}$ (w×l×h) and 4.8 kg. **(b)**: Sketch of the image side telecentric optical setup of the IFPICS prototype. Incident radiation is parallelized by an entrance aperture and lens 1 before traversing the FPI and the band pass filter (BPF). The maximum half cone opening angle $\omega_c$ is dependent on the aperture diameter $a$ and the focal length $f$ of lens 1. The camera field of view is $\theta_{FOV} = 18°$. A second lens maps the image onto a 2-D UV sensitive CMOS detector. **(c)**: Photograph of the static air-spaced etalon (FPI) provided by *SLS Optics Ltd.*. **(d)**: Sketch of the FPI. An incoming single beam (drawn red) with incidence angle $\alpha$ is reflected multiple times between the FPI mirrors with reflectance $R$ and separation $d$. Visualisation of an incoming cone shaped beam (red dash-dotted) with half cone opening angle $\omega_c$ and incidence angle $\alpha$ of the cone axis.

is reached by a small tilt of $\alpha_A = 4.5°$. The basic advantages of using small incident angles $\alpha_i$ are, that they keep the spread of the incidence angle distribution $\gamma(\alpha_i, \omega_c)$ (see Section 2.1) low and thereby retain the FPIs effective finesse $F$ high (since

the reflectivity $R$ of the FPI-mirror coating is somewhat dependent on the angle of incidence so is the finesse $F$). This leads to a much weaker blurring of the FPI fringes in the FPI transmission spectrum $T_{FPI,i}^{eff}$ resulting in a higher sensitivity of the instrument (see Fig. 1). With the prototype setup, however, we encountered disturbing reflections for low FPI incidence angles. For that reason we used the subsequent correlating order of the FPI transmission with $\alpha_A = 8.17°$ for an on-band and $\alpha_B = 6.45°$ for an off-band setting (see Tab. 1), thereby making a compromise between sensitivity and accurate evaluable

images.

**Table 1.** Parameters of the optical components installed in the IFPICS prototype and used in the calibration model. The uncertainties of the model input parameters are shown.

| parameter | value | uncertainties | description |
|---|---|---|---|
| $d\,[\mu\mathrm{m}]$ | 21.666 | $\pm\,0.002$ | FPI plate separation |
| $R$ | 0.65 | | FPI reflectivity |
| $F$ | 7.15 | | FPI finesse |
| $n$ | 1.0003 | | refractive index (air) |
| $\alpha_A\,[°]$ | 8.17* | $\pm\,0.02$ | FPI tilt, on-band |
| $\alpha_B\,[°]$ | 6.45* | $\pm\,0.02$ | FPI tilt, off-band |
| $T_{BPF,max}$ | 0.63 | | BPF peak transmission |
| $\lambda_{BPF}\,[\mathrm{nm}]$ | 308.5 | | BPF central wavelength |
| $\delta_{BPF}\,[\mathrm{nm}]$ | 9.0 | | BPF FWHM |
| $f\,[\mathrm{mm}]$ | 47$^\dagger$ | $\pm\,2$ | lens 1 focal length |
| $a\,[\mathrm{mm}]$ | 1.55 | $\pm\,0.05$ | aperture diameter |
| $\omega_c\,[°]$ | 0.945 | | half cone opening angle |
| $\theta_{FOV}\,[°]$ | 18 | | imaging FOV |

*: used in units of radian in the instrument model Eq. 6 & 7

$^\dagger$: two lenses: $f = \frac{f_1 \cdot f_2}{f_1 + f_2}$ with $f_1 = f_2 \approx 94\,\mathrm{mm}$ @ $\lambda = 310\,\mathrm{nm}$

## 3 Proof of concept study

### 3.1 Measurements at Mt. Etna, Italy

First measurements with the prototype described above were performed at the Osservatorio Vulcanologico Pizzi Deneri (lat 37.766, long 15.017, 2800 m a.s.l.) at Mt. Etna, on July 21 and 22, 2019. The physical properties of the IFPICS prototype and the FPI tilt angles $\alpha_i$ for tuning $T_{FPI,i}^{eff}(\lambda)$ between on-band $i = A$ and off-band setting $i = B$ were selected according to Tab. 1. The tilt of the FPI generates a linear shift between the recorded on-band and off-band images on the detector and accounts for 6 pixel using tilt angles $\alpha_i$. This shift needs to be corrected before cross evaluating images recorded in setting A and B. The exposure time was set to 1 s for all measurements and $4 \times 4$ binning (total spatial resolution of $512 \times 512$ pixels) was applied for all acquired images.

### 3.2 Validation of the instrument model

To quantify the accuracy of our model, two $SO_2$ gas cells were measured with the IFPICS prototype and by Differential Optical Absorption Spectroscopy (DOAS, see Platt and Stutz, 2008), on July 21, 2019, 11:10 - 11:20 CET. The sky was used as light source with a constant viewing angle (10° elevation, 270°N azimuth) in a plume free part of the sky. To enhance the image quality, a flat-field correction is used, compensating pixel to pixel variations in sensitivity. The flat-field correction requires the acquisition of dark and flat-field images. The dark images are determined by the arithmetic mean over five images with no light

entering the IFPICS instrument and the flat-field images are obtained by the arithmetic mean over five images acquired in a plume free sky region. The flat field images thereby directly include the reference measurement $I_{0,i}$, making a later correction for the atmospheric background unnecessary. In the same viewing direction $I_i$ is measured for each gas cell and FPI setting $i$ in order to calculate the AA according to Eq. 1. Figure 4 shows the gas cell measurements (red) including uncertainties (error-bars, 1-$\sigma$). The uncertainties directly arise from the errors of the DOAS measurement and due to variations in optomechanical settings of the IFPICS prototype.

The instrument model (Eq. 2 - 7) was used to calculate the IFPICSs AA $\tilde{\tau}_{SO_2}(S_{SO_2})$ from a given $SO_2$ CD $S_{SO_2}$. The model parameters are mostly fixed by the IFPICS prototype optics as given in Tab. 1. The remaining parameter, the atmospheric $O_3$ slant column density $S_{O_3}$ (see Eq. 2) is calculated in a geometric approximation $S_{O_3} = VCD_{O_3}/\cos(SZA)$ using the solar zentih angle (SZA) and vertical $O_3$ column density ($VCD_{O_3}$) which both are location, date and time dependent. They were: $SZA = (53 \pm 3)°$ (according to the solar geometry calculator by NOAA) and $VCD_{O_3} = (335 \pm 5)\,DU$ (according to TEMIS database; Veefkind et al., 2006). The $VCD_{O_3}$ can be treated to be approximately constant over the period of a day.

The output of the instrument model (drawn, black) for an SZA of $53°$ is shown in Fig. 4. The model uncertainty (shaded grey) is determined by a root mean square over the errors in the output by individually varying the input parameters within their stated uncertainties. The thus calculated calibration function using the instrument model matches the $SO_2$ gas cells validation measurement within the range of confidence. The model nicely describes the flattening of the AA-CD relation for high CDs (up to $\approx 2.5 \times 10^{18}\,molec\,cm^{-2}$), which originates from the CD dependence of $\Delta\tilde{\sigma}(S)$ (see Eq. 1).

To show the impacts of the SZA on the instrument model the model output is also calculated for three other SZAs while keeping the other parameters constant. The model output is shown in Fig. 4 for an SZA of $80°$ (dash-dotted grey) for early morning/late afternoon conditions, SZA of $70°$ (dashed grey) for morning/afternoon conditions and SZA of $25°$ (dotted grey) for noon conditions. High SZAs lead to an increase of stratospheric $O_3$ absorption which alters the spectral shape of the scattered skylight radiance $I_0(\lambda)$ (see Eq. 2) which is used in the forward model. In other words, for high $O_3$ absorption, lower wavelength radiance, where the differential $SO_2$ absorption features are stronger, will contribute less to the integrated radiances $I_i$, $I_{0,i}$ (Eq. 3, 4). The thereby induced SZA dependence of the sensitivity can easily be accounted for in the model. Note that this influence of strong $O_3$ absorption only occurs at our chosen wavelength range for the $SO_2$ measurement. When applying IFPICS to other trace gases, e.g. $BrO$ or $NO_2$ at higher wavelength, this effect will be negligible.

### 3.3 Results of the field measurements

Volcanic plume measurements were performed on July 22, 2019, 08:50 - 09:10 CET. The instrument was pointing towards the plume of Mt. Etna's South East crater with a constant viewing direction (azimuth $204°N$, elevation $5°$, see Fig. 5). The wind direction was $\approx 5°N$ with a velocity of $\approx 6\,m\,s^{-1}$ (wind data from UWYO). Hence, the plume was partly covered by the crater flank. The frame rate during the measurement was $0.2\,Hz$ for a pair ($I_A$ and $I_B$) of images.

The flat-field correction was performed as described in section 3.2, using the arithmetic mean over ten dark images and five flat-field images, obtained in a plume free sky region. An exemplary set of volcanic plume $SO_2$ images, obtained with the IFPICS instrument in on-band setting $I_A$ and off-band setting $I_B$, are shown in Fig. 6. Further images of $I_A$ and $I_B$ are shown

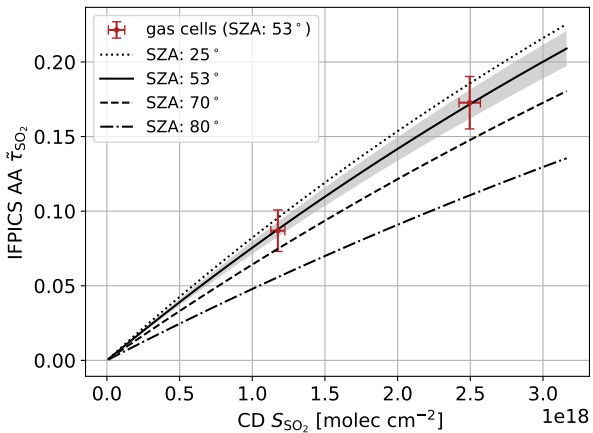

**Figure 4.** The validation measurement with two $SO_2$ gas cells (red, with 1-$\sigma$ error) with the IFPICS prototype and by DOAS on 21 July 2019, 11:10 - 11:20 CET with a solar zenith angle (SZA) of $53°$.

The instrument forward model (Eq. 2-7) is used to calculate the IFPICS AA $\tilde{\tau}_{SO_2}$ for a given CD $S_{SO_2}$ range. The model input parameters are shown in Tab. 1 and $(335\pm5)\,DU$ is used as $VCD_{O_3}$. The calculated model output (black) is shown for four different SZAs ($25°$ (dotted), $53°$ (drawn), $70°$ (dashed) and $80°$ (dash-dotted)). The model output and the validation measurement are in good agreement if a model SZA of $53°$ is used, which is equivalent to the SZA during the measurement time. The model uncertainty is shown in shaded grey.

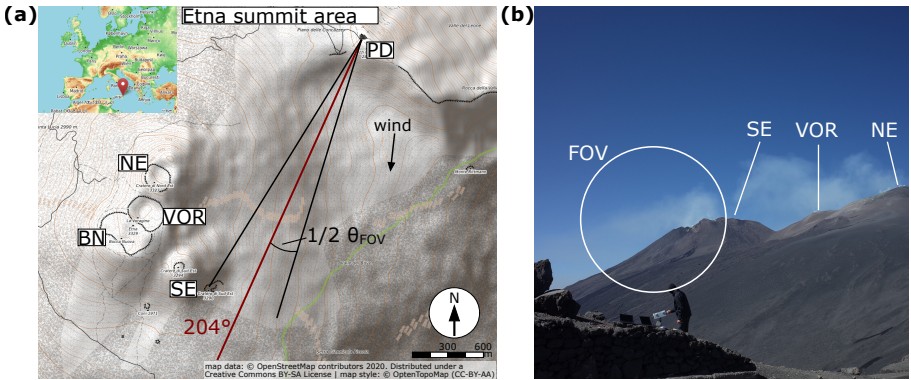

**Figure 5. (a)**: Topographic map of the Mt. Etna summit area, North East crater (NE), Voragine (VOR), Bocca Nuova (BN), South East crater (SE) and measurement location at the Osservatorio Vulcanologico Pizzi Deneri (PD) are indicated. The viewing direction on 22 July 2019 is $204°$ (red drawn) with an FOV of $\theta_{FOV} = 18°$ (black drawn) and an elevation of $5°$. The FOV is partly covering the plume emanating from SE crater. The average wind direction is $\approx 5°$ with a speed of $\approx 6\,m\,s^{-1}$ (wind data from UWYO). **(b)**: Visual image of the volcanic plume on 22 July 2019 with indicated camera field of view (FOV).

in Appendix A. The circular shape of the retrieved image arises from the FPI's circular clear aperture limiting the imaging FOV.

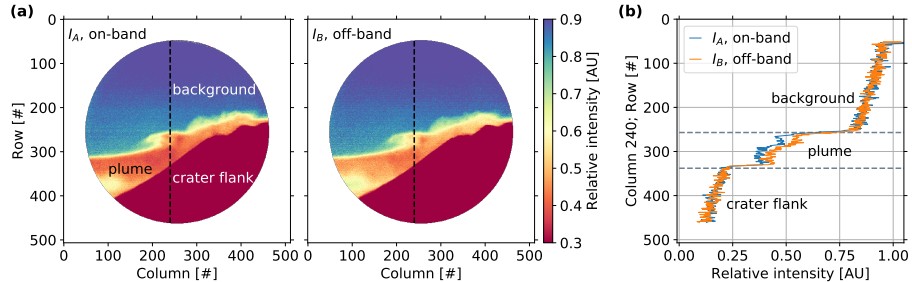

**Figure 6. (a)**: Flat-field corrected intensity images ($400 \times 400$ pixel) acquired with the IFPICS prototype in on-band $I_A$ and off-band setting $I_B$. The $I_A$ image shows the expected higher $SO_2$ absorption in comparison with $I_B$ ($I_A < I_B$ in plume region). The plume is visible in both images due to the broad band $SO_2$ absorption and other extinction in the measurement spectral range. The circular image shape arise from the FPIs circular clear aperture. **(b)**: Intensity column 240 (dashed black lines in (a)) for $I_A$ (blue) and $I_B$ (orange). The enhanced absorption (reduced intensity) is clearly visible in the plume section with $I_A < I_B$, whereas in the background sky and crater flank sections the intensities are equal $I_A = I_B$.

The IFPICS $SO_2$ AA $\tilde{\tau}_{SO_2}$ is calculated pixel-wise according to Eq. 1 from $I_A$ and $I_B$. For the conversion into $SO_2$ CD $S_{SO_2}$ the forward instrument model (Eq. 2 - 7) is inverted by least square fitting of a 4th order polynomial to the calculated CD relation $S_{SO_2}(\tilde{\tau}_{SO_2})$. The model input parameters of the instrument are shown in Tab. 1. The SZA during the time of the measurement is $(78 \pm 3)°$ (NOAA) with a $VCD_{O_3}$ of $335 \pm 5\,DU$ (according to TEMIS database; Veefkind et al., 2006). The retrieved calibration function $S_{SO_2}(\tilde{\tau}_{SO_2})$ is

$$S_{SO_2}(\tilde{\tau}_{SO_2}) = \sum_0^4 x_i \cdot \tilde{\tau}_{SO_2}^i \tag{8}$$

with $x_0 \overset{!}{=} 0$, $x_1 = 1.8 \times 10^{19}$, $x_2 = 1.7 \times 10^{19}$, $x_3 = 1.7 \times 10^{19}$, and $x_4 = 6.6 \times 10^{19}$ in units of $molec\,cm^{-2}$ respectively with $x_0$ fixed to zero. This approximation yields an average relative deviation of $0.007\,\%$ for $S_{SO_2}$ from the modelled value, with a maximum relative deviation of $0.08\,\%$ for small $SO_2$ CDs.

An evaluated image of the volcanic plume $SO_2$ CD distribution corresponding to the intensities shown in Fig. 6 is shown in Fig. 7. Further evaluated CD distribution images of the same time series are presented in Appendix A. A time series of the plume evolution is visualised in a flip-book from the supplementary material.

The volcanic plume of Mt. Etna's South East crater is clearly visibly and reaches $SO_2$ CDs higher than $3 \times 10^{18}\,molec\,cm^{-2}$. The atmospheric background is $S_{SO_2,bg} = 4.3 \times 10^{16}\,molec\,cm^{-2}$ and was determined by the arithmetic mean over a plume free area within the evaluated image (white square, $100 \times 100$ pixel, in Fig. 7, (a)). Since the $S_{SO_2,bg}$ is determined from an evaluated CD distribution image it accounts for the residual signal in $S_{SO2}$ between the direction of the volcanic plume and the direction of the flat-field images used in the evaluation. The $S_{SO_2,bg}$ was subtracted from the displayed image in the final step of the evaluation. The similar plume free area (white square, $100 \times 100$ pixel, in Fig. 7, (a)) is further used to give an estimation for the $SO_2$ detection limit of the IFPICS prototype by calculating the 1-$\sigma$ pixel-pixel standard deviation. The

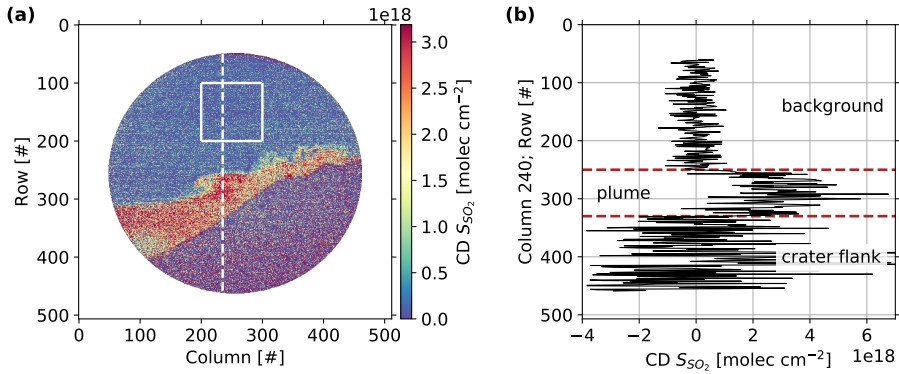

**Figure 7. (a)**: Volcanic plume $SO_2$ CD distribution calculated from images acquired with the IFPICS prototype and using the instrument forward model conversion function $S_{SO_2}(\tilde{\tau}_{SO_2})$ (see Eq. 8). The plume free area indicated by a white square ($100 \times 100$ pixel) is used to correct for atmospheric background and to obtain an estimation for the detection limit. **(b)**: Individual $SO_2$ CD column 240 (indicated by dashed white line in (a)) showing that background, plume, and crater flank region are clearly distinguishable. High scattering in the crater flank region is induced by low radiance.

obtained detection limit for an exposure time of one second is $5.5 \times 10^{17}\,\mathrm{molec\,cm^{-2}}$ given by the noise equivalent signal. The measurements were performed in the morning with an SZA of $78°$ and therefore reduced sensitivity and under relatively low light conditions. For decreasing SZA the sensitivity will increase according to Fig. 4 and the increasing sky radiance will
reduce the photon shot noise. In other words, the gas cell measurements (taken at SZA of $53°$, with approximately twice the sky radiance compared to SZA of $78°$) show a detection limit of $2.1 \times 10^{17}\,\mathrm{molec\,cm^{-2}}$ for an exposure time of one second. For ideal measurement conditions (lowest SZA, highest sky radiance) the detection limit will be further improved.

After the proof of concept, showing the capability of IFPICS to determine $SO_2$ CD images it is possible to determine fluxes from a CD image time series. Especially, if the series allows to trace back individual features in consecutively recorded images
it can be used to directly determine the plume velocity using the approach of cross-correlation (e.g. McGonigle et al., 2005; Mori and Burton, 2006; Dekemper et al., 2016) or optical flow algorithms (e.g. Kern et al., 2015b) and to determine the plume propagation direction (e.g. Klein et al., 2017). However, the viewing geometry on the day of our measurement was unfavourable as it was not possible to reach another measurement location due to a lack in infrastructure. The plume propagation direction and central line of sight show an inclination of $19°$ only, resulting in high pixel contortions, especially for pixel close to the
edges of the FOV. Further, significant parts of the plume are covered by the crater flank due to its propagation direction. For the sake of completeness, we would like to give a rough estimate on the $SO_2$ flux obtained from our data.

The $SO_2$ flux $\Phi_{SO_2}$ is determined by integrating the $SO_2$ CD along a transect through the volcanic plume and subsequent multiplication by the wind velocity perpendicular to the FOV direction, however due to the viewing geometry issues we will use external wind data (direction: $5°$; velocity $v_{wind} \approx 6\,\mathrm{m\,s^{-1}}$ (data from UWYO)) for the calculation. As the camera pixel

size is finite the integral is replaced by a discrete summation over the pixel $n$

$$\Phi_{SO_2} = v_\perp \sum_n S_{SO_2,n} \cdot h_n \tag{9}$$

including the perpendicular wind velocity $v_\perp$, the $SO_2$ CD $S_{SO_2,n}$ and the pixel extent $h_n$. The perpendicular wind velocity can directly be calculated from geometric considerations (see Fig. 5,(a)), accounting to $v_\perp \approx \sin(19°) \cdot v_{wind} \approx 2\,\mathrm{m\,s^{-1}}$. To determine the pixel extent the distance between the volcanic plume and the location of measurement is required. In the centre

of the FOV this distance is $\approx 3500\,\mathrm{m}$ yielding $h_n \approx 2.7\,\mathrm{m}$. To keep the impact of pixel contortions low the plume transect is located centrally in the FOV at column 250 and ranging from rows $n = 230$ to 330. Using these quantities, we retrieve a mean $SO_2$ mass flux for the measurement of $\Phi_{SO_2} = (84 \pm 11)\,\mathrm{t\,d^{-1}}$ for the investigated plume of the South East crater, which is comparable to previous flux measurements of the South East crater (Aiuppa et al., 2008; D'Aleo et al., 2016). Nevertheless, the flux should be regarded as lower limit, since the plume was covered by crater flank to an unknown extent.

## 4  Conclusion

By imaging and quantifying the $SO_2$ distribution in the volcanic plume of Mt. Etna we successfully demonstrate the feasibility of the IFPICS technique proposed by Kuhn et al. (2014). We were able to unequivocally resolve the dynamical evolution of $SO_2$ in a volcanic plume with a high spatial and temporal resolution ($400 \times 400$ pixel, 1 s integration time, $4 \times 4$ binning). The retrieved detection limit for the $SO_2$ measurement is $5.5 \times 10^{17}\,\mathrm{molec\,cm^{-2}\,s^{-1/2}}$. The detection limit however varies with the

SZA and can reach values below $2 \times 10^{17}\,\mathrm{molec\,cm^{-2}\,s^{-1/2}}$ under ideal conditions, comparable to traditional $SO_2$ imaging techniques (see Kern et al., 2015a). Also, the imaging technique lends itself to the determination of gas fluxes and we obtained an $SO_2$ mass flux of $\Phi_{SO_2} = (84 \pm 11)\,\mathrm{t\,d^{-1}}$ for Mt. Etna's South East crater plume. However, due to unfavourable conditions in the viewing geometry the retrieved flux should be treated as a lower limit. In general, it is possible to apply optical flow algorithms on image series acquired under more ideal viewing geometry conditions (e.g. Kern et al., 2015b). These allow to

determine the plume velocity and angle between the observation direction and plume propagation direction in order to retrieve accurate $SO_2$ fluxes (e.g. Klein et al., 2017).

The specific spectral detection scheme of IFPICS allows to use a numerical instrument model to directly convert the measured AA $\tilde{\tau}$ into CD $S$ distributions. This inherent calibration method makes in-field calibrations methods, e.g. by gas cells, unnecessary. The accuracy of the instrument model could be demonstrated using $SO_2$ cells with a known CD, determined by

simultaneous DOAS measurements.

Our IFPICS instrument is still an early stage prototype. The employed optics are highly modular allowing easy adjustments even outside a laboratory. The physical dimensions of $< 10$ litres, and $< 5\,\mathrm{kg}$ and the low power consumption of $< 10\,\mathrm{W}$ combined with the fact that no maintenance and in-field calibration is needed, make it already a close to ideal field instrument. Furthermore, the temporal resolution of the instrument can further be increased by replacing the employed sensor as it does

not allow for time-optimised control of image acquisition.

Compared to traditional $SO_2$ cameras the minimised cross interferences to broad band plume extinction increases the selec-

tivity and thus should allow to apply the IFPICS technique to much weaker $SO_2$ sources. Furthermore, the expected smaller interference to broadband effects in comparison to traditional $SO_2$ imaging techniques should allow to extend the range of meteorological conditions acceptable for field measurement (see Kuhn et al., 2014).

The demonstrated IFPICS technique is not limited to the detection of $SO_2$. In general the technique is applicable to numerous further trace gases which show a distinct pattern (ideally periodic) in their absorption spectrum (see Kuhn et al., 2019). In the case of volcanic emissions detectable trace species are e.g. bromine monoxide $BrO$ or chlorine dioxide $OClO$. Beyond volcanic applications IFPICS could be used to investigate e.g. air pollution by measuring nitrogen dioxide $NO_2$ or formaldehyde $HCHO$.

**Appendix A**

Further evaluated images of the time series acquired on July 22, 2019, 08:50 - 09:10 CET at Mt. Etna, Italy are shown in Fig. A1. The evaluation procedure is analogous to the routine explained in Section 3.3. The time difference between a set (1 - 4) of images accounts for $\approx 120\,\mathrm{s}$ and allows to trace back plume dynamics.

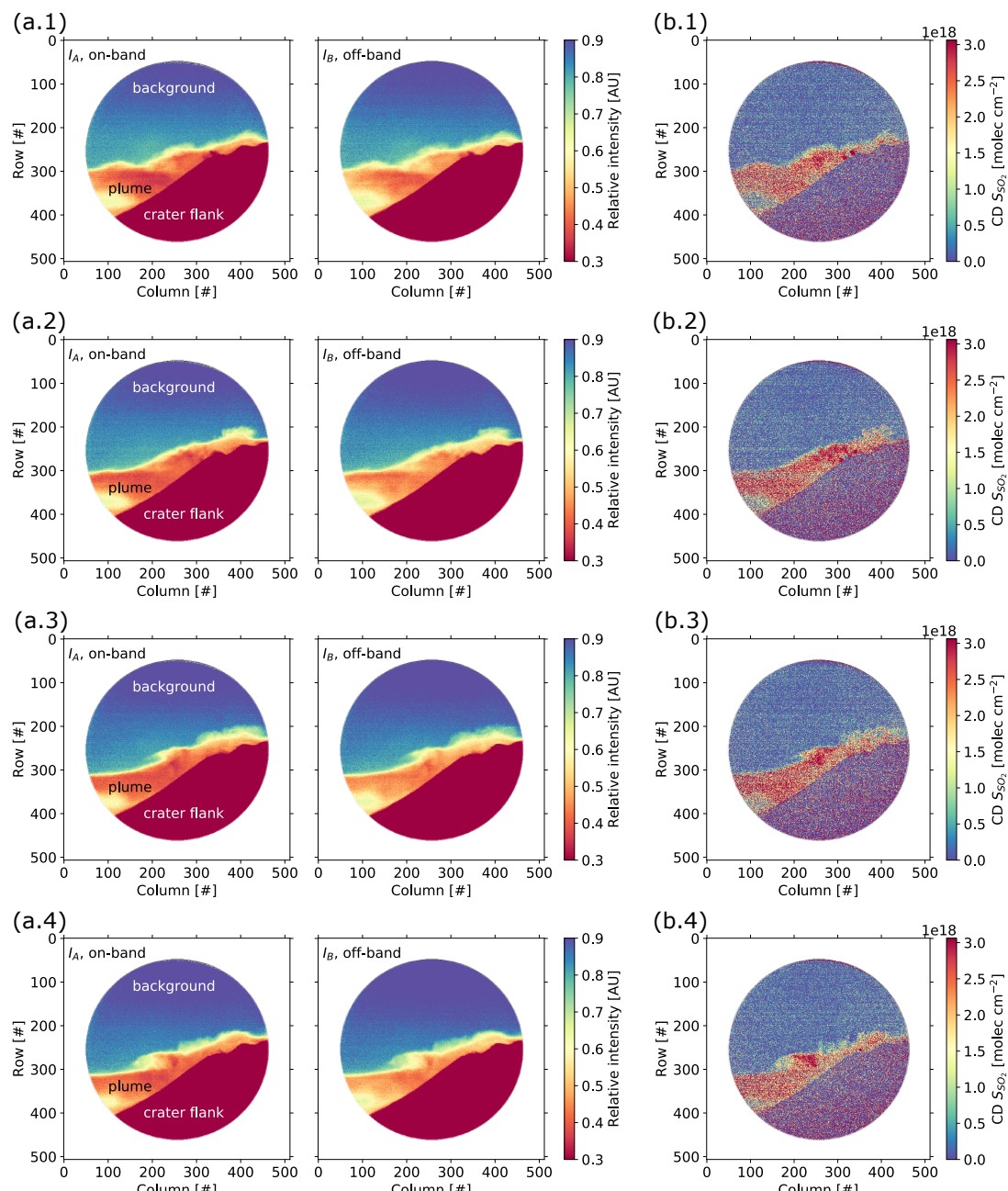

**Figure A1.** Exemplary set of evaluated images (400×400 pixel) acquired with the IFPICS prototype on July, 22, 2019, 08:50 - 09:10 CET at Mt. Etna, Italy. The time difference between each set of images (**1-4**) accounts for $\approx 120\,\mathrm{s}$, allowing to trace back plume dynamics. **(a)**: Flat-field corrected intensity images $I_A$ and $I_B$. **(b)**: Volcanic plume $SO_2$ CD $S_{SO_2}$ distribution calculated with the conversion function shown in Eq. 8.

*Data availability.* The data can be obtained from the authors upon request.

*Author contributions.* JK, NB and UP developed the question of research. JK, NB and CF conducted the field campaign. JK and CF developed the instrument model. CF designed, constructed and characterised the instrument, evaluated the data and wrote the manuscript with all authors contributing by revising it within several iterations.

*Competing interests.* The authors declare that they have no conflict of interest.

*Acknowledgements.* We would like to thank *SLS Optics Ltd.* for sharing their expertise in designing and manufacturing the etalons. Support by the Deutsche Forschungsgemeinschaft (project DFG PL 193/23-1) is gratefully acknowledged. We also thank Emmanuel Dekemper and Toshiya Mori for their valuable and constructive reviews.

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
