# Peer review of "Quantitative imaging of volcanic SO2 plumes with Fabry Pérot Interferometer Correlation Spectroscopy"

_Atmospheric Measurement Techniques, 2020_

## Referee Comment (RC1) · Emmanuel Dekemper (Referee) · 11 Aug 2020

**1   General evaluation:**

The manuscript describes a major development step for this very elegant atmospheric measurement technique already initiated several years ago. The idea of matching the transmission comb of a Fabry-Pérot interferometer (FPI) with the regular structures present in the absorption spectrum of the target atmospheric species finds a very convincing application here with the remote sensing of volcanic SO2. The instrument concept, and the field campaign are well described, and my feeling is that the overall

quality (both in terms of content and language) is already quite high with this initial submission. Though, I have a few points of concerns which I would like to raise. They are discussed below.

**1.1 The abstract.**

Although the abstract is a good summary of the manuscript (high level description of the instrument concept, and the experimental results achieved), I think it is slightly exaggerating the demonstrated capabilities of the instrument. For instance, it is claimed that the instrument does the job for SO2, BrO, and NO2, whereas only the first species is addressed. I understand that the prototype was designed to correlate with the SO2 structures, but therefore, at least a theoretical simulation of performance for the other species should have been presented. In absence of this, the BrO and NO2 capabilities should only be referred to as potential future applications. The same goes for the statement that the instrument allows to determine gas fluxes, while this aspect is also not discussed in the paper. The factual performance of the prototype is also a bit misleading: the claimed integration time of 1s is, as far as I could understand, the integration time of a single image, not yet the temporal resolution of the geophysical product (presumably closer to 5 seconds) as it seems currently suggested. Hence, I would recommend to rework a bit the abstract such that undemonstrated, though potentially achievable goals are not presented as conlusions of the work.

**1.2 The instrumental model.**

The mathematics describing the measurements have been carefully developed, and the reader will appreciate the author's will to integrate all the meaningful aspects of the model (in particular the splitting of the instrument transfer function into different multiplicative terms). However, I have two remarks regarding this section:

1. Less experienced readers might be lost in this section because it lacks a drawing representing the light paths involved. Supporting the mathematical description with a figure showing that eq.(2) refers to the light path originating from the Sun and going up to the point of scattering into the instrument line of sight would already be helpful. Having two rays illustrating the difference between $I_i$, and $I_{0,i}$ would also be appreciated.

2. Recalling the reader about the fundamental FPI equation is valuable. However, eq.(7) appears to be a step too far, especially that the weighting function term N remains mysterious at the end. I believe that the discussion on the effective transmission spectrum of the FPI is an important point. But because the reader will anyway not be able to reproduce your model (because of the undetermined term N), it is better to illustrate the effect of increasing the acceptance angle (or the tilt angle) on the FPI transmission with the help of a figure (a bit like fig.(1), but emphasizing the change of $T_{FPI}$ as a function of these angles). Also, I found it not so clearly explained that the way the comb of the FPI is shifted (to go from setting A to B and back) is by rotating the FPI axis. A few words about the different means of performing this shift with nowadays FPI technologies (e.g. MEMS, piezo), and the trade off which led to the selection of the tilting approach would be appreciated.

**1.3  Minor comments.**

- p.2,l.26: The NO2 camera, presented in Dekemper et al. 2016, has a spectral resolution of 0.6nm at 440nm... The statement that native spectral imagers have a "strongly reduced spectral resolution" is therefore not correct. It is not because the classical filter-based SO2 cameras have a poor spectral resolution that all other spectral imagers have the same drawback, especially when the filter technology is completely different.

- I was wondering if the tilting of the FPI in order to go from setting A to B was introducing a shift of the respective images onto the detector? Is there a re-alignment step needed in the pre-processing of the data? If yes, then this is worth a couple of sentences addressing this aspect.

- Section 3.2: Your forward model uses a geometric air mass factor to estimate the SCD of O3. The model was validated for a relatively small SZA with the two gas cells. However, your field measurements were performed with a much larger SZA of almost 80°. Don't you expect a bias coming from the geometric AMF in that circumstances?

- p.10,l.203: How did you estimate the background SO2? Your method relies on using the background signal in order to determine the CD in the plume. Which I0 did you use for the determination of the background SO2?

**1.4 Typos.**

- p.4,l.80: stratosperhic -> stratospheric

- p.7,l.145: describe -> described

- p.7,l.151: add a comma after "model"

- p.7,l.154: add a comma after "quality"

- p.7,l.157: including -> include

- p.8,l.173: add a comma after the first "model"

- On several occasions, the form "I. e." is used at the beginning of a sentence (like on p.8, line 177). I don't understand this abbreviation.

- p.8,l.179: remove the comma after "Note"

- p.10,l.199: start a new paragraph with "An evaluated ..."

- p.11,l.229: "increases selectivity" -> "increases the selectivity"

---

## Referee Comment (RC2) · Toshiya MORI (Referee) · 2 Sep 2020

This manuscript presents a sophisticated imaging technique using an interferometer (FPI) for volcanic SO$_2$. A developed prototype instrument, IFPICS solves several issues of the conventional SO$_2$ cameras which use broad interference filters. The technical background of the newly developed instruments is well described and the results of the field observation using the prototype IFPICS seems very promising for the future application in volcano monitoring. I have several comments on the manuscript as shown below.

[Figure]

Specific comments:

"2.2 The IFPICS prototype"

From Figure 2, it seems that the tilt angles of FPI is controlled by a stepping motor. However, there seems to be no description on how the tilt angles for the two settings A and B are controlled in the manuscript. Although the optics of the IFPICS are explained in detail, the mechanical part of the IFPICS is poorly explained. The mechanical part of the IFPICS prototype especially about the changing of the tilt angle should also be described in the manuscript. How long does it take to change the tilt angle? This may partly explain rather low frame rate of 0.2 Hz for the pair of images.

"Table 1 and Equation 6"

Direct use of the parameter values in Table 1 into equation 6 seems inappropriate. Either the values in Table1 or the equation 6 needed to be modified. The sine in eq. 6 is in radiance and the cosine is in degree. They should be matched. d and $\lambda$ in eq. 6 needed to be in the same unit or conversion factor should be included in eq 6.

"Figure 6"

In Fig.6(b), CD $S_{SO2}$ value between Row 400 and 415 (most part is hidden behind the "crater flank" label) seems to be shifted to positive side unlike those of other Rows (distributed around zero). As stated in the end of the figure caption of Fig. 5, $I_A$ is basically equal to $I_B$ for both background sky and flank. Is there any possibility of $SO_2$ on the flank or is there any other reason to explain for the positive shift? According to a Global Volcanism Program report in "Global Volcanism Program, 2019. Report on Etna (Italy) (Crafford, A.E., and Venzke, E., eds.). Bulletin of the Global Volcanism Network, 44:10. Smithsonian Institution. https://doi.org/10.5479/si.GVP.BGVN201910-211060." There was a lava flow event between 19-21 July, 2019 (until a day before the observation) on the eastern flank of SEC.

Probably part of the flow may have been in the view of the IFPICS at the time of the observation on 22 July, 2019. Is there any possibility detecting volcanic fume with $SO_2$ from the lava flow?

Lines 194-200:" The SZA during the time of the measurement is (78±3) (NOAA) with a $VCDO_3$ retrieved calibration function ...with x0 = $1.0\times10^{13}$, x1 = $1.1\times10^{19}$, x2 = $9.3\times10^{18}$, x3 = $7.9\times10^{18}$, and x4 = $1.6\times10^{19}$ in units of molec cm$^{-2}$ respectively."

Reading here and the figure caption of Fig. 6, x0-x4 parameters is supposed to correspond to SZA=78 degrees. As I plotted Eq. 8 with x0-x4 values, it seems the conversion function correspond to SZA 25 degrees. Please give the parameters for SZA=78 degrees corresponding to the observation. If the conversion function with x0-x4 given in the manuscript are used for calculation of $SO_2$ CD distributions in Fig. 6, $SO_2$ CD need to be recalculated using appropriate conversion function.

"Equation 8"

According to equation 1, AA is zero, if $SO_2$ CD (S) is zero. Considering this, 0th order parameter x0 may not be needed or may be set to zero in the 4th order polynomial fitting.

Line 203:" The atmospheric background is $S_{SO2,bg}$ = 4.3×$10^{16}$ molec cm$^{-2}$"

Definition of atmospheric background $S_{SO2}$ is not clear. Does this value correspond to the difference of $S_{SO2}$ between plume direction and flat-field image direction or to the absolute atmospheric background value for observation direction?

Lines 230-231: "Furthermore, the small interference to broadband effects extends the range of meteorological conditions acceptable for field measurement."

I agree that one of the major advantages of the IFPICS is extension of acceptable meteorological ranges in the field measurements such as minimal influence of background clouds. I suppose the author need to explain more specific on this. Personally, I feel slightly pity because the authors did not show clear example images corresponding to outcome of "the small interference to broadband effects" in this manuscript, which would definitely convince the readers of the clear advantages of the new IFPICS compared to the conventional $SO_2$ cameras.

Minor comments:

Line 190: "The circular shape of the retrieved image arises from the FPI's circular clear aperture limiting the imaging FOV." And, line 216:" a high spatial and temporal resolution (400×400 pixel, 1 s integration time)"

The 2D UV-sensitive CMOS sensor (SCM2020-UV) is originally a 2000x2000 pixels

sensor. It seems 4x4 pixel binning is applied to the images. If so, please indicate in the manuscript.

Lines 205-207:" The similar plume free area (white square, 100 $\times$ 100 pixel, in Fig. 6, (a)) is further used to give an estimation for the $SO_2$ detection limit of the IFPICS prototype by calculating the 1-$\sigma$ pixel-pixel standard deviation. The obtained detection limit is $5.5 \times 10^{17}$ molec cm$^{-2}$ s$^{-\frac{1}{2}}$ given by the noise equivalent signal."

Please explain how the detection limit was calculated more in detail. 1-sigma pixel-pixel standard deviation does not seem to give the detection limit unit indicated here.

Figure caption of Fig. A1:" acquired with the IFPICS prototype on 22. July 2019, 08:50 - 09:10 CET"

Delete "." after "on 22"

Other comment:

It would be helpful, especially for non-volcanological readers, to show visual image of the plume from the observation site if available.

———————————————

---

## Author Comment (AC1) · 16 Nov 2020

**Author's response to Review RC1 by Emmanuel Dekemper**

We are very gratefully acknowledging the comments of Emmanuel Dekemper. The comments are highly helpful in both, in enhancing the clarity of the presented technique and model.

For clarity we answer the specific comments directly (bold printed). The reviewer comments are set in italic font, the authors' responses in normal font. We added a new Figure (Fig. 2) and included a SO2 flux calculation at the end of Section 3.3 and in Section 4. In several places throughout the manuscript we modified and extended sentences yielding minor changes to the manuscript.

**1.1 The abstract**

_Reviewer's comment_: _Although the abstract is a good summary of the manuscript (high level description of the instrument concept, and the experimental results achieved), I think it is slightly ex-aggerating the demonstrated capabilities of the instrument. For instance, it is claimed that the instrument does the job for SO2, BrO, and NO2, whereas only the first species is addressed. I understand that the prototype was designed to correlate with the SO2 structures, but therefore, at least a theoretical simulation of performance for the other species should have been presented. In absence of this, the BrO and NO2 capabilities should only be referred to as potential future applications. The same goes for the statement that the instrument allows to determine gas fluxes, while this aspect is also not discussed in the paper. The factual performance of the prototype is also a bit misleading: the claimed integration time of 1s is, as far as I could understand, the integration time of a single image, not yet the temporal resolution of the geophysical product (presumably closer to 5 seconds) as it seems currently suggested. Hence, I would recommend to rework a bit the abstract such that undemonstrated, though potentially achievable goals are not presented as conclusions of the work._

Author's response:
- Indeed, we did only present imaging measurement results of volcanic $SO_2$ emissions. Of course, the FPI employed for that was specifically designed to correlate with $SO_2$. However, the camera prototype itself was not solely implemented for $SO_2$. Replacing the BPF and the FPI, by one that is designed for BrO or $NO_2$ the camera can be used for measuring further gas species. Within this manuscript we did in fact not present theoretical simulation of the performance for gases other than $SO_2$ since these were already presented in a former manuscript (Kuhn et al. 2019). We will therefore change the claim to be able to measure BrO and NO2 to future applications.

  We changed the sentence (submitted manuscript lines: 4 - 6):
  "Matching the FPIs distinct, periodic transmission features to the characteristic differential absorption structures of the investigated trace gas allows to measure differential atmospheric column density (CD) distributions of numerous trace gases, e.g. sulphur dioxide ($SO_2$), bromine monoxide (BrO), or nitrogen dioxide ($NO_2$), with high spatial and temporal resolution."

To (revised manuscript lines: 3 - 6):
"Matching the FPIs distinct, periodic transmission features to the characteristic differential absorption structures of the investigated trace gas allows to measure differential atmospheric column density (CD) distributions of numerous trace gases with high spatial and temporal resolution. Here we demonstrate measurements of sulphur dioxide ($SO_2$) while earlier model calculations show that bromine monoxide (BrO) and nitrogen dioxide ($NO_2$) are also possible."

- Further, we stated that we can determine gas fluxes since it is usually possible to retrieve fluxes from an image time series with sufficient high spatial resolution. However, the viewing geometry on the day of measurement was quite unfavourable as the plume propagation direction and field of view direction only had low inclination of 19° in contrast to ideal condition with a plume perpendicular to the viewing direction. Also, the fact that the plume was partly covered by the crater flank only allows to calculate a lower limit to the actual gas flux. Nonetheless, we now include a flux calculation in the article and discussing the unfavourable conditions for this particular example, which lead to a rather high measurement error but is showing the capability of IFPICS to determine fluxes.

  We added the sentences (revised manuscript lines: 238 - 259):
  "After the proof of concept, showing the capability of IFPICS to determine $SO_2$ CD images it is possible to determine fluxes from a CD image time series. Especially, if the series allows to trace back individual features in consecutively recorded images it can be used to directly determine the plume velocity using the approach of cross-correlation (e.g. McGonigle et al., 2005; Mori and Burton, 2006; Dekemper et al., 2016) or optical flow algorithms (e.g. Kern et al., 2015b) and to determine the plume propagation direction (e.g. Klein et al., 2017). However, the viewing geometry on the day of our measurement was unfavourable as it was not possible to reach another measurement location due to a lack in infrastructure. The plume propagation direction and central line of sight show an inclination of 19° only, resulting in high pixel contortions, especially for pixel close to the edges of the FOV. Further, significant parts of the plume are covered by the crater flank due to its propagation direction. For the sake of completeness, we would like to give a rough estimate on the $SO_2$ flux obtained from our data.
  The $SO_2$ flux $\Phi_{SO2}$ is determined by integrating the $SO_2$ CD along a transect through the volcanic plume and subsequent multiplication by the wind velocity perpendicular to the FOV direction, however due to the viewing geometry issues we will use external wind data (direction: 5°; velocity $v_{wind} \approx 6$ m s$^{-1}$ (data from UWYO)) for the calculation. As the camera pixel size is finite the integral is replaced by a discrete summation over the pixel $n$

  $$\Phi_{SO2} = v\perp \sum_n S_{SO2;n} \cdot h_n$$

  including the perpendicular wind velocity $v\perp$, the SO2 CD $S_{SO2;n}$ and the pixel extent $h_n$. The perpendicular wind velocity can directly be calculated from geometric considerations (see Fig. 5, (a)), accounting to $v\perp \approx sin(19°) \, v_{wind} \approx 2$ m s$^{-1}$. To determine the pixel extent the distance between the volcanic plume and the location of measurement is required. In the centre of the FOV this

distance is $\approx$3500m yielding $h_n \approx$2,7 m. To keep the impact of pixel contortions low the plume transect is located centrally in the FOV at column 250 and ranging from rows n = 230 to 330. Using these quantities, we retrieve a mean SO$_2$ mass flux for the measurement of $\Phi_{SO2}$ = (84±11) t d$^{-1}$ for the investigated plume of the South East crater. Nevertheless, the flux should be regarded as lower limit, since the plume was covered by crater flank to an unknown extent."

We changed & relocated the sentence (submitted manuscript lines: 231 - 233): "Also, the imaging technique lends itself to the determination of gas fluxes. For instance, the wind velocity and also the angle between the observation direction and plume propagation direction can be determined from the image series."

To (revised manuscript lines: 266 - 271): "Also, the imaging technique lends itself to the determination of gas fluxes and we obtained an SO2 mass flux of $\Phi_{SO2}$ = (84±11) t d$^{-1}$ for Mt. Etna's South East crater plume. However, due to unfavourable conditions in the viewing geometry the retrieved flux should be treated as a lower limit. In general, it is possible to apply optical flow algorithms on image series acquired under more ideal viewing geometry conditions (e.g. Kern et al., 2015b). These allow to determine the plume velocity and angle between the observation direction and plume propagation direction in order to retrieve accurate so2 fluxes (e.g. Klein et al., 2017)."

- We added the total time required for the acquisition of a pair of images, including tilting, and saving the images, which was 5.5 seconds for the used prototype setup. In further instrument versions, image readout and motor movement are negligible compared to the exposure time of 1 s.

  We further changed the sentence (submitted manuscript lines: 9 - 11): "In a field campaign, we recorded the temporal CD evolution of SO2 in the volcanic plume of Mt. Etna with an integration time of 1 s and 400x400 pixels spatial resolution. The first IFPICS prototype can reach a detection limit of $2,1 \times 10^{17}$ molec cm$^{-2}$ s$^{-1}$, which is comparable to traditional and much less selective volcanic SO$_2$ imaging techniques."

  To (revised manuscript lines: 9 - 14): "In a field campaign, we recorded the temporal CD evolution of SO$_2$ in the volcanic plume of Mt. Etna with an exposure time of 1 s per image and 400 x 400 pixel spatial resolution. The temporal resolution of the time series was limited by the available non-ideal prototype hardware to about 5.5 s. Nevertheless, a detection limit of $2,1 \times 10^{17}$ molec cm$^{-2}$ could be reached, which is comparable to traditional and much less selective volcanic SO2 imaging techniques."

**1.2 The instrumental model**

*Reviewer's comment: The mathematics describing the measurements have been carefully developed, and the reader will appreciate the author's will to integrate all the meaningful aspects of the model (in particular the splitting of the instrument transfer function into different multiplicative terms). However, I have two remarks regarding this section:*

1. *Less experienced readers might be lost in this section because it lacks a drawing representing the light paths involved. Supporting the mathematical description with a figure showing that eq.(2) refers to the light path originating from the Sun and going up to the point of scattering into the instrument line of sight would already be helpful. Having two rays illustrating the difference between Ii, and I0,i would also be appreciated.*

2. *Recalling the reader about the fundamental FPI equation is valuable. However, eq.(7) appears to be a step too far, especially that the weighting function term N remains mysterious at the end. I believe that the discussion on the effective transmission spectrum of the FPI is an important point. But because the reader will anyway not be able to reproduce your model (because of the undetermined term N), it is better to illustrate the effect of increasing the acceptance angle (or the tilt angle) on the FPI transmission with the help of a figure (a bit like fig.(1),but emphasizing the change of $T_{FPI}$ as a function of these angles). Also, I found it not so clearly explained that the way the comb of the FPI is shifted (to go from setting A to B and back) is by rotating the FPI axis. A few words about the different means of performing this shift with nowadays FPI technologies (e.g. MEMS, piezo), and the trade off which led to the selection of the tilting approach would be appreciated.*

Author's response:
**Point 1:** This is a good suggestion. We extended the introduction of the mathematical model by a graphical representation, which is now Fig. 2.
> "A 2D UV-sensitive CMOS sensor (SCM2020-UV provided by *EHD imaging*) is used to acquire images."

We included a new figure; Fig. 2:

[Figure]

**Figure 2.** Schematic of the IFPICS measurement geometry including the simulated radiances used in the instrument model. The incident top of atmosphere (TOA) radiation $I_{0,TOA}$ is propagating through the atmosphere and is potentially scattered into the IFPICS camera field of view (FOV) yielding the scattered skylight radiance $I_0$. The camera records radiation in the respective FPI settings i = A and B that either traverses the volcanic plume $I_i$ or originates from a plume free area within the FOV $I_{0,i}$.

**Point 2:** We had many thoughts about how detailed we should present our applied model. We tried to make it as detailed as possible and the representation of the final equation (Eq. (7)) was frequently discussed among the authors. Due to its high complexity, requiring three case analyses, all resulting in a different function, we ultimately only showed a more general equation trying to emphasize the basic principle. The quantity $N$ representing the weighting and is given by the integral in Eq. 7 excluding the integrand $T_{FPI,i}$. We will include a description in the revised version of the manuscript.

Thanks for the comment concerning the shift of the FPI comb. We will emphasize this point in both, the introduction of the model, and in the description of the prototype setup. Further we will add a description to the manuscript why we use a tilting approach.

We added the sentences (revised manuscript lines: 104 - 107):
"The FPI used in this work is static and air-spaced, meaning $d$, $n$, and $R$ are fixed. Hence, the incidence angle $\alpha_i$ is the exclusive free parameter available to tune the FPIs transmission spectrum $T_{FPI;i}$ between settings $i = A$ and $i = B$ respectively. The change in $\alpha_i$ is achieved by tilting the FPI optical axis with respect to the imaging optical axis (see Section 2.2)."

We changed the sentence (submitted manuscript lines: 107 - 108):
"Thereby, $N(\gamma(\alpha_i,\omega_c))$ denotes the weighting function, $\vartheta$ the polar angle and $\varphi$ the azimuth angle of the spherical integration within boundaries defined by the tilted cone shaped light beams."

To (revised manuscript lines: 115 - 117):
"Thereby, $N(\gamma(\alpha_i,\omega_c))$ denotes the weighting function with $N(\gamma(\alpha_i,\omega_c)) = \iint \sin\vartheta \, d\vartheta \, d\varphi$ given by the integral in Eq. 7 excluding the integrand $T_{FPI,I}$ itself, $\vartheta$ the polar angle and $\varphi$ the azimuth angle of the spherical integration within boundaries defined by the tilted cone shaped light beams."

We changed and extended the sentence (submitted manuscript lines: 125 - 127):
The static air-spaced FPI ($d$, $n$ and $R$ fixed, provided by *SLS Optics Ltd.*) can be tilted within the parallelised light path in order to tune its spectral transmission $T^{eff}_{FPI}$ between setting A and B via variation of the incidence angle $\alpha$ (see Section 2.1).

To (revised manuscript lines: 135 - 138):
The FPI is the central optical element of the IFPICS prototype and is implemented as static air-spaced etalon with fixed $d$, $n$, and $R$ (provided by *SLS Optics Ltd.*). The mirrors are separated using ultra low expansion glass spacers to maintain a constant mirror separation $d$ and parallelism over the large clear aperture of 20 mm even under highly variable environmental conditions. In order to tune the spectral transmission $T^{eff}_{FPI}$ between setting A and B a variation of the incidence angle $\alpha$ is applied.

We added the sentence (revised manuscript lines: 141 - 143):
We favour the approach of tilting the FPI over changing internal physical properties like, e.g. the mirror separation $d$ by piezoelectric actuators, as it keeps simplicity, robustness, and accuracy high for measurements under non-laboratory conditions.

**1.3 Minor comments**

- *Reviewer's comment: p.2,l.26: The NO2 camera, presented in Dekemper et al. 2016, has a spectral resolution of 0.6nm at 440nm... The statement that native spectral imagers have a "strongly reduced spectral resolution" is therefore not correct. It is not because the classical filter-based SO2 cameras have a poor spectral resolution that all other spectral imagers have the same drawback, especially when the filter technology is completely different.*

  Author's response:

  We changed and extended the sentence (submitted manuscript lines: 24 - 30):
  "A third approach applies a small number of (typically two) wavelength channels by using wavelength selective optical elements for the entire image frame, thereby usually strongly reducing the spectral resolution (e.g. Mori and Burton, 2006; Dekemper et al., 2016). The high spectral resolution of the first two, spectrograph based approaches allows the accurate and simultaneous identification of several trace gases, however, the light throughput and the scanning process severely limit the temporal resolution. The third approach can be quite fast, the trace gas selectivity, however, strongly depends on the correlation of trace gas absorption with the wavelength selective elements and usually is rather marginal."

  To (revised manuscript lines: 26 - 34):
  "The high spectral resolution of the spectrograph based techniques allows the accurate and simultaneous identification of several trace gases, however, the light throughput and the scanning process severely limit the temporal resolution. A third approach applies tunable filters to resolve the trace gas spectral features, e.g. acousto-optical tunable filter (Dekemper et al., 2016), as wavelength selective elements for an entire image frame. The application of tunable filters can have high spectral resolution and hence high trace gas selectivity, however, due to limited light throughput the temporal resolution lies in the order of minutes. A fourth imaging technique uses a small number (typically two) wavelength channels selected by static filters, e.g. interference filters (Mori and Burton, 2006). This approach can be quite fast with a temporal resolution in the order of seconds, the trace gas selectivity, however, strongly depends on the correlation of trace gas absorption with the wavelength selective elements and usually is rather marginal."

- *Reviewer's comment: I was wondering if the tilting of the FPI in order to go from setting A to B was introducing a shift of the respective images onto the detector? Is there a re-alignment step needed in the pre-processing of the data? If yes, then this is worth a couple of sentences addressing this aspect.*

  Author's response: Yes, indeed the tilting is inducing a linear shift of the respective images A and B on the detector. Therefore, a realignment step for the processing is performed accounting for a shift of 6 pixels. In the revised version we will address this point in Section 2.2 and 3.1.

We added the sentence (revised manuscript lines: 143 - 145):
"However it need to be considered, that the tilting of the FPI will generate a linear shift between the respective images acquired in setting A and B, requiring an alignment in the evaluation process."

We added the sentence (revised manuscript lines: 166 - 167):
"The tilt of the FPI generates a linear shift between the recorded on-band and off-band images on the detector and accounts for 6 pixel using tilt angles $\alpha_i$. This shift needs to be corrected before cross evaluating images recorded in setting A and B."

- *Reviewer's comment: Section 3.2: Your forward model uses a geometric air mass factor to estimate the SCD of O3. The model was validated for a relatively small SZA with the two gas cells. However, your field measurements were performed with a much larger SZA of almost 80°. Don't you expect a bias coming from the geometric AMF in that circumstances?*

  Author's response: Yes indeed, large SZA can impact the calibration function retrieved by the instrument model. We geometrically recalculated the $O_3$ AMF assuming a homogeneous spherical shell of $O_3$ within a spherical nonrefracting atmosphere. For an SZA of 78° the retrieved change in the $O_3$ AMF is -7.5% translating to a sensitivity increase of +3.6% of the calculated calibration function. As the bias is rather small, we will not include the correction in model applied in this work. We will extend our model by a more general AMF calculation in future studies.

- *Reviewer's comment: p.10,l.203: How did you estimate the background SO2? Your method relies on using the background signal in order to determine the CD in the plume. Which I0 did you use for the determination of the background SO2?*

  Author's response: The $SO_2$ background signal has no significant impact on the measurement. A plume-free region within the measurement image is used to calculate the differential $SO_2$ signal induced by the plume (see: revised manuscript lines: 226 - 230). The model is not impacted by the atmospheric $SO_2$ background, since its absorption does not significantly impact the shape of the solar spectrum in the measurement wavelength range.

**1.4 Typos**

- p.4,l.80: stratosperhic -> stratospheric
  corrected as proposed

- p.7,l.145: describe -> described
  corrected as proposed

- p.7,l.151: add a comma after "model"
  corrected as proposed

- p.7,l.154: add a comma after "quality"

corrected as proposed

- p.7,l.157: including -> include
corrected as proposed

- p.8,l.173: add a comma after the first "model"
corrected as proposed

- On several occasions, the form "I. e." is used at the beginning of a sentence (like on p.8, line 177). I don't understand this abbreviation.
We changed the abbreviation "I.e." occurring at the beginning of the sentences either into "In other words" or "That is to say".

- p.8,l.179: remove the comma after "Note"
corrected as proposed

- p.10,l.199: start a new paragraph with "An evaluated ..."
corrected as proposed

- p.11,l.229: "increases selectivity" -> "increases the selectivity"
corrected as proposed

---

## Author Comment (AC2)

**Author's response to Review RC2 by Toshiya Mori**

First, we like to gratefully thank for the constructive, detailed and helpful comments given by Toshiya Mori. We are convinced that the comments allowed us to improve the manuscript quality within the revision process.

For clarity we answer the specific comments directly (bold printed). The reviewer comment is set in italic font, the authors response in normal font. We added a figure in section 3.3 (Fig. 5 (b)), recalculated the coefficients of Eq. 8, and recalculated the Fig. 7 with new values of Eq. 8. On several occasions we changed and added sentences resulting in minor changes to the manuscript.

**"2.2 The IFPICS prototype"**

Reviewer's comment: From Figure 2, it seems that the tilt angles of FPI is controlled by a stepping motor. However, there seems to be no description on how the tilt angles for the two settings A and B are controlled in the manuscript. Although the optics of the IFPICS are explained in detail, the mechanical part of the IFPICS is poorly explained. The mechanical part of the IFPICS prototype especially about the changing of the tilt angle should also be described in the manuscript. How long does it take to change the tilt angle? This may partly explain rather low frame rate of 0.2 Hz for the pair of images.

Author's response: This is a valid comment. In the revised version we include the mechanical description of the IFPICS prototype:

The tilt angle for the two settings A and B is - as mentioned in the comment - controlled by a stepping motor. The motor has a step resolution of 0.9°. It is equipped with an additional planetary gearbox with a reduction ratio of 1 to 9 reducing the effective step resolution to 0.1°. The motor is controlled by a microcontroller combined with a stepping motor controller. The controller enables the operation of the stepper motor in micro-stepping mode thus further improving the angular resolution by a factor of 16, yielding a final resolution of 0.00625 degrees per motor step. An optical switch is used for position sensing of the stepper motor.

The time required for changing the tilt from setting A to setting B is of the order of 0.15 s. Hence, the low frame rate of the prototype of 0.2 Hz (5.5 seconds per pair of frames) mainly arises from controlling and triggering the employed UV detector.

We changed and extended the sentence: (submitted manuscript lines: 125 - 127): "The static air-spaced FPI (*d*, *n* and *R* fixed, provided by *SLS Optics Ltd*.) can be tilted within the parallelised light path in order to tune its spectral transmission  $T^{\text{eff}}_{FPI}$  between setting A and B via variation of the incidence angle  $\alpha$  (see Section 2.1)."

**To (revised manuscript lines: 135 - 138):**

"The FPI is the central optical element of the IFPICS prototype and is implemented as static air-spaced etalon with fixed *d*, *n*, and *R* (provided by *SLS Optics Ltd*.). The mirrors are separated using ultra low expansion glass spacers to maintain a constant mirror separation *d* and parallelism over the large clear aperture of 20 mm even in variable environmental conditions. In order to tune the spectral transmission  $T^{eff}_{FPI}$  between setting A and B a variation of the incidence angle  $\alpha$  is applied."

We added the sentence: (revised manuscript lines: 138 - 141):

"The FPI can be tilted within the parallelised light path using a stepper motor. The stepper motor has a resolution of 0.9° per step, is equipped with a planetary gearbox (reduction rate 1/9) and operated in micro-stepping mode (1/16) resulting in a resolution of 0.00625° per motor step. The time required for tilting between our settings A and B is  $\approx$  0.15 s."

**Table 1 and Equation 6**

Reviewer's comment: Direct use of the parameter values in Table 1 into equation 6 seems inappropriate. Either the values in Table 1 or the equation 6 needed to be modified. The sine in eq. 6 is in radiance and the cosine is in degree. They should be matched. d and  $\lambda$  in eq. 6 needed to be in the same unit or conversion factor should be included in eq. 6.

Author's response: Thanks for that comment. We will include a note for the units required for sine and cosine calculation. For d and  $\lambda$  the units of  $\mu$ m and nm are indicated in Tab. 1.

We added the footnote to Tab. 1: "\*: used in units of radian in the instrument model Eq. 6 & 7"

**Figure 6**

Reviewer's comment: In Fig.6(b), CD SSO2 value between Row 400 and 415 (most part is hidden behind the "crater flank" label) seems to be shifted to positive side unlike those of other Rows (distributed around zero). As stated in the end of the figure caption of Fig. 5, IA is basically equal to IB for both background sky and flank. Is there any possibility of SO2 on the flank or is there any other reason to explain for the positive shift? According to a Global Volcanism Program report in "Global Volcanism Program, 2019. Report on Etna (Italy) (Crafford, A.E., and Venzke, E., eds.). Bulletin of the Global Volcanism Network, 44:10. Smithsonian Institution. https://doi.org/10.5479/si.GVP.BGVN201910-211060." There was a lava flow event between 19-21 July, 2019 (until a day before the observation) on the eastern flank of SEC. Probably part of the flow may have been in the view of the IFPICS at the time of the observation on 22 July, 2019. Is there any possibility detecting volcanic fume with SO2 from the lava flow?

Author's response: Yes, indeed there was a lava flow event at Mt. Etna close to the time we were measuring. However, we do not expect to detect its fume. The enhanced signal can most likely be explained by the low level of light scattered from the crater flank and the thereby increasing influence of hardware systematics of the detector and statistical fluctuations.

Lines 194-200:" The SZA during the time of the measurement is (78±3) (NOAA) witha VCDO3retrieved calibration function...with  $x0 = 1.0 \times 10^{13}$ ,  $x1 = 1.1 \times 10^{19}$ ,  $x2 = 9.3 \times 10^{18}$ ,  $x3 = 7.9 \times 10^{18}$ , and  $x4 = 1.6 \times 10^{19}$ in units of molec cm-2 respectively." Reviewer's comment: Reading here and the figure caption of Fig. 6, x0-x4 parameters is supposed to correspond to SZA=78 degrees. As I plotted Eq. 8 with x0-x4 values, it seems the conversion function correspond to SZA 25 degrees. Please give the

parameters for SZA=78 degrees corresponding to the observation. If the conversion function withx0-x4 given in the manuscript are used for calculation of SO2 CD distributions in Fig.6, SO2CD need to be recalculated using appropriate conversion function.

Author's response: Yes, the given polynomial parameters accidentally corresponded to an SZA of 25 degrees. However, it was only a copy error of from the model code values into the manuscript. All calculations were performed with the correct calibration function as given in the following:

 $x0 = 1.04 \times 10^{13}$ ,  $x1 = 1.81 \times 10^{19}$ ,  $x2 = 1.73 \times 10^{19}$ ,  $x3 = 1.69 \times 10^{18}$ , and  $x4 = 6.77 \times 10^{19}$ .

**Equation 8**

Reviewer's comment: According to equation 1, AA is zero, if SO2 CD (S) is zero. Considering this,0th order parameter x0 may not be needed or may be set to zero in the 4th order polynomial fitting.

Author's response: Yes, that is true as we retrieve the calibration function from a numerical model. We changed our polynomial fitting using a y-intercept fixed to zero. The new calibration function reads:

x0 = 0,  $x1 = 1.81 \times 10^{19}$ ,  $x2 = 1.72 \times 10^{19}$ ,  $x3 = 1.73 \times 10^{19}$ , and  $x4 = 6.64 \times 10^{19}$ .

We recalculated Fig. 6 (now Fig. 7) using the new calibration function. The changes are marginal and slightly visible in the noise of the crater flank in Fig. 7 (b).

We changed the sentence (submitted manuscript line: 197 - 198): "[...] with  $x0 = 1.0 \times 10^{13}$ ,  $x1 = 1.1 \times 10^{19}$ ,  $x2 = 9.3 \times 10^{19}$ ,  $x3 = 7.9 \times 10^{18}$ , and  $x4 = 1.6 \times 10^{19}$  in units of molec cm-2 respectively."

To (revised manuscript lines: 219 - 220):

"[...] with x0 = 0, x1 =  $1.8 \times 10^{19}$ , x2 = $1.7 \times 10^{19}$ , x3 =  $1.7 \times 10^{19}$ , and x4 =  $6.6 \times 10^{19}$  in units of molec cm-2 respectively with x0 fixed to zero."

We recalculated Fig. 6 (revised manuscript Fig. 7):

**Figure 7. (a):** Volcanic plume SO2 CD distribution calculated from images acquired with the IFPICS prototype and using the instrument forward model conversion function  $S_{SO2}(\tilde{\tau})$  (see Eq. 8). The plume free area indicated by a white square (100 x 100 pixel) is used to correct for atmospheric background and to obtain an estimation for the detection limit. (b): Individual SO2 CD column 240 (indicated by dashed white line in (a)) showing that background, plume, and crater flank region are clearly distinguishable. High scattering in the crater flank region is induced by low radiance.

**Line 203:" The atmospheric background is SSO2,bg= 4.3×1016molec cm-2"**

Reviewer's comment: Definition of atmospheric background  $S_{SO2}$  is not clear. Does this value correspond to the difference of  $S_{SO2}$  between plume direction and flat-field image direction or to the absolute atmospheric background value for observation direction?

Author's response: Thanks for that comment, we shall clarify the definition of the atmospheric background SSO2,bg in the revised manuscript:

The  $S_{SO2,bg}$  accounts for the difference in  $S_{SO2}$  between the plume direction and flatfield image direction.

We added the sentence (revised manuscript lines: 227 - 229):

"Since the  $S_{SO2,bg}$  is determined from an evaluated CD distribution image it accounts for the residual signal in  $S_{SO2}$  between the direction of the volcanic plume and the direction of the flat-field images used in the evaluation."

**Lines 230-231: "Furthermore, the small interference to broadband effects extends the range of meteorological conditions acceptable for field measurement.**

Reviewer's comment: I agree that one of the major advantages of the IFPICS is extension of acceptable meteorological ranges in the field measurements such as minimal influence of background clouds. I suppose the author need to explain more specific on this. Personally, I feel slightly pity because the authors did not show clear example images corresponding to outcome of "the small interference to broadband effects" in this manuscript, which would definitely convince the readers of the clear advantages of the new IFPICS compared to the conventional SO2 cameras.

Author's response: Yes, this is a valid comment. We do expect a weaker influence on broadband interferences for example induced by background clouds in contrast to traditional filter based SO2 cameras. This statement is based on model calculations as shown in Kuhn et al., 2014. However, within this work our main goal was to demonstrate the feasibility of the IFPICS technique to detect volcanic SO2 emissions. Hence, in our up to now limited data set we only took data under good weather conditions without background sky clouds. This fact limits our dataset, and unfortunately, we cannot provide exemplary images yet. Despite that fact, we certainly plan to address this topic in near future studies. For this manuscript we will weaken our statement on this topic.

We changed the sentence (submitted manuscript lines: 230 - 231):

"Furthermore, the small interference to broadband effects extends the range of meteorological conditions acceptable for field measurement."

**To (revised manuscript lines: 282 - 284):**

"Furthermore, the expected smaller interference to broadband effects in comparison to traditional SO2 imaging techniques should allow to extend the range of meteorological conditions acceptable for field measurement (see Kuhn et al., 2014). "

**Minor comments:**

• Line 190: "The circular shape of the retrieved image arises from the FPI's circular clear aperture limiting the imaging FOV." And line 216:" a high spatial and temporal resolution (400×400 pixel, 1 s integration time)" Reviewer's comment: The 2D UV-sensitive CMOS sensor (SCM2020-UV) is originally a 2000x2000 pixels sensor. It seems 4x4 pixel binning is applied to the images. If so, please indicate in the manuscript.

Author's response: Yes, indeed. We applied 4x4 pixel binning and will include this information in the revised version of the manuscript.

We changed the sentence (submitted manuscript lines: 117 - 118): "A 2D UV-sensitive CMOS sensor (SCM2020-UV provided by *EHD imaging*) is used to acquire images."

To (revised manuscript lines: 125 - 126): "A 2D UV-sensitive CMOS sensor (SCM2020-UV provided by *EHD imaging*) is used to acquire images. The sensor is operated in 4x4 binning mode yielding a final image resolution of 512x512pixel."

We changed the sentence (submitted manuscript lines: 215 - 216): "We were able to unequivocally resolve the dynamical evolution of SO2 in a volcanic plume with a high spatial and temporal resolution (400x400 pixel, 1 s integration time)."

To: (revised manuscript lines: 262 - 263):

"We were able to unequivocally resolve the dynamical evolution of SO2 in a volcanic plume with a high spatial and temporal resolution (400x400 pixel, 1 s integration time, 4x4 binning)."

We changed the sentence (submitted manuscript line: 149): "The exposure time was set to 1 s for all acquired images."

To (revised manuscript lines: 167 - 169):

"The exposure time was set to 1 s for all measurements and 4x4 binning (total spatial resolution of 512x512 pixels) was applied to all acquired images."

 Lines 205-207:" The similar plume free area (white square, 100×100 pixel, in Fig.6, (a)) is further used to give an estimation for the SO2 detection limit of the IFPICS prototype by calculating the 1-σ pixel-pixel standard deviation. The obtained detection limit is 5.5×1017molec cm-2s-1/2given by the noise equivalent signal."

Reviewer's comment: Please explain how the detection limit was calculated more in detail. 1-sigma pixel-pixel standard deviation does not seem to give the detection limit unit indicated here.

Author's response: Thank you for this comment. We used the 100x100 pixel area and calculated the respective standard deviation for these pixel. This yields the stated detection limit of  $5.5 \times 10^{17}$  molec cm-2. The unit of s-1/2 arises from the

time dependency of the photon shot noise which is proportional to 1/sqrt(t) with the exposure time t. As the images have been obtained with an exposure time of 1s the unit of s-1/2 can be included. We will clarify this statement in the revised manuscript:

We changed the sentence (submitted manuscript line: 207): "The obtained detection limit is  $5.5 \times 10^{17}$ molec cm-2 s-1/2 given by the noise equivalent signal."

To (revised manuscript lines: 231 - 232):

"The obtained detection limit for an exposure time of one second is  $5.5 \times 10^{17}$  molec cm-2 given by the noise equivalent signal."

 Figure caption of Fig. A1:" acquired with the IFPICS prototype on 22. July 2019, 08:50- 09:10 CET" Delete "." after "on 22" Corrected as proposed

**Other comment:**

Reviewer's comment: It would be helpful, especially for non-volcanological readers, to show visual image of the plume from the observation site if available.

Author's response: We added a visual image to Fig. 4 (revised manuscript: Fig. 5).